# Composition Characterization of *Crossostephium chinense* Leaf Essential Oil and Its Anti-Inflammatory Activity Mechanisms

**DOI:** 10.3390/plants13172506

**Published:** 2024-09-06

**Authors:** Chia-Hsin Lin, Yu-Ting Chiang, Li-Yin Lin, Nai-Wen Tsao, Chung-Hsuan Wang, Shih-Chang Chien, Ying-Hsuan Sun, Sheng-Yang Wang

**Affiliations:** 1Department of Forestry, National Chung Hsing University, Taichung 402202, Taiwan; jocelyn0625@hotmail.com (C.-H.L.); z0987719303@gmail.com (Y.-T.C.); yhsun@email.nchu.edu.tw (Y.-H.S.); 2Department of Chinese Pharmaceutical Sciences and Chinese Medicine Resources, China Medical University, Taichung 404328, Taiwan; 3Liyu International Co., Ltd., Taichung 402202, Taiwan; liliant0720@gmail.com; 4Special Crop and Metabolome Discipline Cluster, Academy Circle Economy, National Chung Hsing University, Taichung 402202, Taiwan; n.w.tsao.qq@gmail.com (N.-W.T.); piscium0312@smail.nchu.edu.tw (C.-H.W.); 5Experimental Forest Management Office, National Chung-Hsing University, Taichung 402202, Taiwan; scchien@nchu.edu.tw; 6Agricultural Biotechnology Research Center, Academia Sinica, Taipei City 115201, Taiwan

**Keywords:** *Crossostephium chinens*, essential oil, inflammation, NRF2/HO-1 antioxidant pathway

## Abstract

This study investigates the composition characteristics and anti-inflammatory activity mechanisms of the essential oil from the leaves of *Crossostephium chinense*. *C. chinense* is a perennial herb commonly found in East Asia, traditionally used to treat various ailments. The essential oil extracted through water distillation, primarily contains 1,8-cineole (13.73%), santolina triene (13.53%), and germacrene D (10.67%). Three compounds were identified from the essential oil, namely 1-acetoxy-2-(2-hydroxypropyl)-5-methylhex-3,5-diene, 1-acetoxy-isopyliden-hex-5-en-4-one, and chrysanthemyl acetate, with the first two being newly discovered compounds. Then, the essential oil of *C. chinense* exhibits significant anti-inflammatory effects on RAW264.7 macrophages, effectively inhibiting the production of NO and ROS, with the IC50 value of 10.3 μg/mL. Furthermore, the essential oil reduces the expression of pro-inflammatory cytokines such as TNF-α, IL-6, and IL-1β. Mechanistic studies indicate that the essential oil affects the inflammatory response by inhibiting the expression of iNOS but has no significant impact on COX-2. Further analysis suggests that the essential oil may regulate the inflammatory response through the ERK protein in the MAPK pathway and IκBα in the NF-κB pathway, while also promoting the activity of the NRF2/HO-1 antioxidant pathway, enhancing the cell’s antioxidant capacity, thereby achieving an effect of inhibiting the inflammatory response. These results highlight the potential application value of *C. chinense* leaf essential oil in the medical and healthcare fields.

## 1. Introduction

*Crossostephium chinense*, commonly known as “Qing Ai” in Chinese, is a perennial herb. It is classified as a plant in the Asteraceae family, which is one of the largest families of flowering plants and includes many other well-known species such as daisies, sunflowers, and chrysanthemums. Currently, *C. chinense* is the only widely recognized and accepted species within the *Crossostephium* genus. This genus is relatively small, and no other species have been commonly accepted or documented in the scientific literature to date. This plant is native to East Asia, particularly China, where it has been used for centuries in traditional medicine and various cultural practices. *C. chinense* typically grows to a height of about 30 to 60 cm. The plant features gray-green, deeply lobed leaves with a soft, downy texture. The leaves are aromatic, releasing a distinctive, pleasant fragrance when crushed. The plant produces small, yellowish flowers in clusters, which bloom from late spring to early summer. In traditional Chinese medicine, *C. chinense* is valued for its therapeutic properties. It is often used to treat a variety of ailments, including digestive disorders, skin conditions, and respiratory issues. The plant is believed to possess anti-inflammatory, antibacterial, antifungal, and anticancer properties [1]. *C. chinense* holds a significant place in Chinese culture.

The plant is often associated with protection and purification. During certain festivals and rituals, branches of the plant are hung in homes or carried as amulets to ward off evil spirits and bring good fortune in Taiwan. Studies have shown that the water crude extract has good in vitro antioxidant activity and can inhibit the proliferation activity of the human liver cancer cell HepG2. It can also achieve hepatoprotective effects by improving the antioxidant capacity in rats [2]. In addition, the ethyl acetate-soluble part and the water-soluble part in the ethanol crude extracts are good inhibitory effects on α-glucosidase activity [3]. It also inhibits the differentiation of osteoclasts and reduces bone resorption, which is helpful for the treatment of gout. The methanol extract can be used to fight lung cancer cells and various parasites, such as those that cause malaria falciparum or the trypanosomes that cause African trypanosomiasis [4]. The essential oil extracted by distillation can kill mosquito larvae [3], and the hydrosol can prevent intimal hyperplasia caused by coronary artery balloon dilation [4].

The components of *C. chinense* were isolated from the ethanol extract of the whole plant, including flavonoids 5,7-dihydroxy-3′,4′,5′-trimethoxyflavonoids (tricetin 3′,4′,5′-trimethylether), hispidulin, 5′,5,7-trihydroxy-3′,4′-dimethoxyflavone (apometzgerin), marigold flavin-3,6,7-tri methyl ether (quercetagetin 3,6,7-trimethylether), 4′,5′,5,7-tetrahydroxy-3′-methoxyflavone (selagin), and quercetin-7-O-β-D-glucoside (quercetin-7-O-β-D-glucoside) [5] and triterpene compounds, including taraxerol, taraxeryl acetate, α-amyrin acetate, β-amyrin acetate, and neooilexonyl acetate. In addition, there are also studies on isolating coumarin components, such as scopoletin, from the ethanol extract [6]. Fifty-six compounds have been identified as volatile components. The compound with the highest proportion is isocaryophllene. Another study mentioned that the main components of the essential oil are santolina triene and 1,8-cineole [7]. Recent studies have performed liquid–liquid partition extraction on the methanol extract to obtain the n-hexane-soluble part, the ethyl acetate-soluble part, the n-butanol-soluble part, and the water-soluble part, and a variety of compounds were isolated from the ethyl acetate-soluble part and the n-butanol-soluble part, including coumarin derivatives, flavonoids, flavonoid glycosides, and caffeic acid derivatives, and many new ψ-santonin compounds were discovered [4].

Anti-inflammatory refers to the property of a substance or treatment that reduces inflammation or swelling in the body. Inflammation is a biological response to harmful stimuli such as pathogens, damaged cells, or irritants and is a protective mechanism intended to remove these harmful stimuli and initiate the healing process. However, excessive or chronic inflammation can lead to various diseases. Anti-inflammatory agents work by counteracting or suppressing the mechanisms that lead to inflammation [8]. Several biochemical pathways are involved in the inflammatory process, and the key pathways include the cyclooxygenase (COX) pathway [9], lipoxygenase (LOX) pathway [10], nuclear factor-kappa B (NF-κB) pathway, mitogen-activated protein kinase (MAPK) pathway [11], and JAK-STAT pathway [12] and pro-inflammatory cytokines like TNF-α, IL-1, IL-6, and IL-8 [13]. Anti-inflammatory agents work through various mechanisms and pathways to reduce or inhibit the body’s inflammatory response. Understanding these pathways helps in the development of targeted therapies for inflammatory diseases, providing better outcomes with fewer side effects [14].

Compared to the crude extracts from various solvents of *C. chinense*, research on the activity and components of essential oils is relatively scarce. This study aims to separate and purify the essential oil, conduct structural identification using spectral analysis, analyze the unknown components, and explore its application prospects in medicine and healthcare through anti-inflammatory research.

## 2. Results and Discussion

### 2.1. Chemical Composition of C. chinense Essential Oil (CCEO)

The essential oil of *C. chinense* leaves was extracted by water distillation. After distillation, the average essential oil yield was calculated to be 0.093% (*v*/*w*).

The major constituents of CCEO were determined via GC/MS analysis, and their relative contents (%) are presented in Table 1. In the essential oil, the prominent compounds include 1,8-cineole (13.73%), santolina triene (13.53%), germacrene D (10.67%), cubebol (5.79%), *trans*-caryophyllene (3.29%), germacrene B (3.6%), and α-cadinol (3.54%). The components of the essential oil in the same plant are various due to different cultivation climates, soils, and geographical environments [15]. According to the research results, isocaryophllene is the most common essential oil of *C. chinense* leaves, while another study pointed out that santolina triene (50.9%) is the main component of CCEO, followed by 1,8-cineole (17.89%) [7], which is very similar to the results of our study. It is speculated that this may be the result of the close geographical location.

There are three new compounds isolated from the essential oil. One of which is chrysanthemyl acetate, and the other two are the new compounds 1-acetoxy-isopyliden-hex-5-en-4-one, accounting for 2.6% of the total components, and 1-acetoxy-2-(2-hydroxypropyl)-5-methylhex-3,5-diene, accounting for 0.23% of the total composition. In addition, the essential oil of *C. chinense* contains some less common compounds that are also found in the essential oils of Asteraceae plants, such as santolina triene, chrysanthemol, artedouglasia oxide, and lanciniata furanone derivatives [16].

### 2.2. New Compounds Come from CCEO by HPLC and NMR

We found that in the GC/MS analysis, the structures of several compounds could not be determined, and there was a lack of standards. Therefore, chromatography and spectroscopic analyses were used to identify and prepare standards. There are three new compounds (Figure 1A–C). (The NMR spectra are shown in the Appendix A).

Compound **1** was obtained using SPE from CCEO and separated with an *n*-hexane/ethyl acetate = 50:50 solvent elution. This fraction was subjected to HPLC, and the portion with a retention time of 10.55 ± 0.18 min was collected. In the ^1^H-NMR spectrum (δ_H_, CDCl3, 400 MHz), signals of the methyl group were observed at δ_H_ 1.03 (3H, s), 1.26 (3H, s), 2.03 (3H, s), and 1.88 (3H, t, *J* = 1.2 Hz); two terminated olefinic protons were present at δ_H_ 5.74 (1H, br, t, *J* = 1.2 Hz) and 5.89 (1H, br, s); two methylene signals were present at δ_H_ 4.03 (1H, dd, *J* = 11.6, 8.0 Hz) and δ_H_ 4.17 (1H, dd, *J* = 11.6, 6.8 Hz). Additionally, methine signals appeared at δ_H_ 2.10 (1H, d, *J* = 5.2 Hz) and δ_H_ 1.93 (1H, ddd, *J* = 11.6, 8.0, 5.2 Hz). Combining ^13^C-NMR (δ_C_, CDCl3, 100 MHz), DEPT, and HSQC spectra reveals that compound **1** contains 12 carbons. The primary carbons are at δ_C_ 17.7, 20.0, 21.0, and 21.1, corresponding to δ_H_ 1.88, 1.03, 2.03, and 1.26, respectively. The secondary carbons are at δ_C_ 63.8 and 124.2, with the former corresponding to δ_H_ 4.03 and 4.17 and the latter to δ_H_ 5.74 and 5.89. The tertiary carbons are at δ_C_ 29.7 and 36.5, corresponding to δ_H_ 1.93 and 2.10, respectively. Finally, the quaternary carbons are at δ_C_ 29.5, 146.1, 171.1, and 198.6. Generally, quaternary carbon signals located after δ_C_ 160 are indicative of carbonyl groups. Therefore, δ_C_ 171 and 198.6 are inferred to be carbonyl groups, while δ_C_ 146.1 is inferred to be a quaternary carbon with an olefinic bond. From the HMBC spectrum, it is observed that δ_H_ 4.17 and 4.03 are correlated with δc 29.7, 36.5, 29.5, and 171.1; δ_H_ 1.93 is correlated with δ_C_ 36.5, 198.6, and 20.0; δ_H_ 2.10 is correlated with δ_C_ 63.8, 29.7, 198.6, 29.5, and 21.1; δ_H_ 1.88 is correlated with δc 198.6, 146.1, and 124.2; δ_H_ 5.89 and 5.74 are correlated with δc 198.6, 146.1, and 17.7; δ_H_ 1.03 is correlated with δ_C_ 29.7, 36.5, 29.5, and 21.1; δ_H_ 1.03 is correlated with δc 29.7, 36.5, 29.5, and 20.0; δ_H_ 2.03 is correlated with δ_C_ 63.8 and 171.1. From the COSY spectrum and coupling constants, it is found that δ_H_ 1.88 is coupled with δ_H_ 5.74, and δ_H_ 1.93 is coupled with δ_H_ 4.17 and 4.03. Based on the above evidence, compound **1** is elucidated as 1-acetoxy-2R,3R-isopropyliden-hex-5-en-4-one, a novel compound that has not been previously reported (Figure 1A).

Compound **2** showed a molecular of C_12_H_20_O_3_ established by EI-HRMS (212.1425). In the ^1^H-NMR spectrum (δ_H_, CDCl3, 400 MHz), 19 hydrogen signals were observed, and methyl signals appeared at δ_H_ 1.20 (two overlapping), 1.82, and 3.05; methylene signals appeared at δ_H_ 4.17 (dd, *J* = 8.4, 11.2 Hz, 1H), δ_H_ 4.28 (dd, *J* = 4.8, 11.2 Hz, 1H), and δ_H_ 4.93; and three methine signals were observed at δ_H_ 2.44, 5.49, and 6.21. Based on ^13^C-NMR, DEPT, and HSQC spectra, it is known that this compound has a total of 12 carbons. The primary carbons are located at δ_C_ 18.6, 21.0, 27.1, and 28.4, corresponding to δ_H_ 1.82, 2.00, 1.20, and 1.20, respectively. The secondary carbons are at δ_C_ 64.7 and 116.2, corresponding to δ_H_ 4.17, 4.28, and δ_H_ 4.93. The tertiary carbons are at δ_C_ 53.3, 126.9, and 137.2, corresponding to δ_H_ 2.44, 5.49, and 6.21. The quaternary carbons are at δ_C_ 71.8, 141.5, and 171.0. The quaternary carbon at δ_C_ 71.8 is inferred to be an oxygen-adjacent carbon, while the carbon at δ_C_ 171.0 is likely a carbonyl group. From the HMBC spectrum, it is observed that δ_H_ 4.17 and 4.28 are correlated with δ_C_ 53.3, 126.9, 137.2, and 71.8; δ_H_ 2.44 is correlated with δ_C_ 64.7, 126.9, 137.2, and 71.8; δH 5.49 is correlated with δ_C_ 64.7, 53.3, and 141.5; δ_H_ 6.21 is correlated with δ_C_ 53.3, 141.5, 116.2, and 18.6; δ_H_ 4.93 is correlated with δ_C_ 137.2, 141.5, and 18.6; δ_H_ 1.82 is correlated with δ_C_ 137.2, 141.5, and 116.2; δ_H_ 1.2 is correlated with δ_C_ 53.3, 71.8, 27.1, or 28.4; δ_H_ 2.00 is correlated with δ_C_ 171.0. From the COSY spectrum and coupling constants, it is known that δ_H_ 1.82 is coupled with δ_H_ 4.93; δ_H_ 2.44 is coupled with δ_H_ 4.28, 4.17, and δ_H_ 5.49; and δ_H_ 5.49 is coupled with δ_H_ 6.21. Based on the above results, this compound is identified as (*E*)-1-acetoxy-2-(2-hydroxypropyl)-5-methylhex-3,5-diene, a newly discovered compound reported here for the first time (Figure 1B).

Compound **3** showed a molecule of C_12_H_20_O_2_ established by EI-HRMS (196.1467) (S8.). The ^1^H-NMR spectrum (δ_H_, CDCl3, 400 MHz) revealed a total of 20 hydrogen signals. The methyl signals appeared at δ_H_ 1.03, 1.10, 1.65, 1.68, and 2.03; methylene signals appeared at δ_H_ 3.96 (dd, *J* = 8.8, 11.6 Hz, 1H) and δ_H_ 4.23 (dd, *J* = 6.8, 7.2 Hz, 1H), and three methine signals were observed at δ_H_ 0.82 (ddd, *J* = 5.2, 6.8, 8.8 Hz, 1H), δ_H_ 1.14 (dd, *J* = 5.2, 7.6 Hz, 1H), and δ_H_ 4.85 (m, 1H). Based on ^13^C-NMR, DEPT, and HSQC spectra, this compound contains 12 carbons. The primary carbons are at δ_C_ 18.2, 21.1, 21.5, 22.4, and 25.6, corresponding to δ_H_ 1.65, 2.03, 1.10, 1.03, and 1.68. The secondary carbon is at δ_C_ 65.6, corresponding to δ_H_ 3.96 and 4.23. The tertiary carbons are at δ_C_ 28.9, 30.9, and 122.9, corresponding to δ_H_ 1.14, 0.82, and 4.85. The quaternary carbons are at δ_C_ 22.3, 133.7, and 171.3. The quaternary carbon at δ_C_ 22.3 is inferred to be connected to four substituents; δ_C_ 133.7 is a C–C double bond, and δ_C_ 171.3 is a carbonyl group. From the HMBC spectrum, it is observed that δ_H_ 3.96 and 4.23 are correlated with δ_C_ 30.9, 28.9, 22.3, and 171.3; δ_H_ 0.82 is correlated with δ_C_ 122.9, 22.3, 21.5, and 22.4; δ_H_ 1.14 is correlated with δ_C_ 65.6 and 133.7; δ_H_ 4.85 is correlated with δ_C_ 133.7, 18.2, and 25.6; δ_H_ 1.65 is correlated with δ_C_ 122.9, 133.7, and 25.6; δ_H_ 1.68 is correlated with δ_C_ 122.9, 133.7, and 18.2; δ_H_ 1.10 is correlated with δ_C_ 30.9, 28.9, and 22.4; δ_H_ 1.03 is correlated with δ_C_ 30.9, 28.9, and 21.5; and δ_H_ 2.03 is correlated with δ_C_ 171.3. From the COSY spectrum and coupling constants, it is found that δ_H_ 4.85 is coupled with δ_H_ 1.14 and 1.65, and δ_H_ 0.82 is coupled with δ_H_ 1.14, 4.23, and 3.96. This compound is chrysanthemyl acetate (Figure 1C).

### 2.3. Anti-Inflammatory Activity of CCEO

The MTT assay was used to evaluate whether CCEO is cytotoxic to RAW264.7 [17]. Compared with the group without drug treatment, the survival rate was above 80% under the treatments with concentrations of 5, 10, and 20 µg/mL. However, at concentrations above 40 µg/mL, the survival rate of RAW264.7 is slightly less than 80%, so the optimal essential oil concentration is controlled within 20 µg/mL for subsequent experiments (Figure 2). The NO inhibition rates treated with CCEO concentrations of 5, 10, and 20 µg/mL were 25.2%, 47.2%, and 81.6% (Figure 3A), respectively, and the half-inhibitory concentration IC_50_ was 10.3 µg/mL. Compared with *Origanum ehrenbergii* essential oil (IC_50_ = 66.37 μg/mL) and *Cinnamomum insularimontanum* fruit essential oil (IC_50_ = 18.68 μg/mL) [18] they have better effects. On the other hand, the ROS in RAW264.7 cells was detected and measured with DCFH-DA. As shown in Figure 3B, the fluorescence intensity of the LPS-induced group increased significantly, indicating that LPS would induce an increase in intracellular ROS concentration and cause oxidative stress, and CCEO treatment at 10 and 20 μg/mL can significantly reduce ROS production. From this result, it is inferred that CCEO can simultaneously inhibit the inflammation and oxidative stress induced by LPS, and has the good anti-inflammatory and antioxidant effects.

### 2.4. Effect of CCEO on Pro-Inflammatory-Related Factors Induced by LPS in RAW264.7 Cells

It is known that when the immune system is activated by factors such as foreign pathogens or tissue damage, it will produce a large number of inflammatory mediators, such as NO, TNF-α, IL-6, IL-1β, and lipid mediators, such as PGs, among which PGE2 is mainly produced by COX-2 in the inflammatory response [19]. As can be seen from Figure 4, RAW264.7 cells activated by LPS produce a large amount of pro-inflammatory cytokines, including TNF-α, IL-6, IL-1β, and COX-2. The gene expressions of IL-1β and IL-1β showed a downward trend as the concentration increased after adding CCEO. At a concentration of 20 μg/mL, it was significantly inhibited. However, the gene expression of COX-2 was not inhibited by CCEO (Figure 4A). In order to further verify the above results, the contents of PGE2, TNF-α, IL-6, and IL-1β in the culture medium of different groups were measured. The ELISA results are shown in Figure 5. There are four pro-inflammatory factors in the culture medium of the LPS group. The contents of TNF-α and IL-6 were significantly decreased in the CCEO treatment groups at different concentrations, while the contents of PGE2 and IL-1β were also significantly decreased in the highest concentration of 20 μg/mL.

### 2.5. Effect of CCEO on iNOS and COX-2 Protein in LPS-Induced RAW264.7 Cells

Since the production of NO in the inflammatory reaction is regulated by iNOS and PGE2 content is mainly regulated by COX-2, the Western blot method was used to detect the expression of three proteins in the cells [19]. The results are shown in Figure 6. The expression of iNOS protein increased significantly after LPS induction treatment with 20 µg/mL. CCEO can significantly reduce the expression of iNOS protein (Figure 6A); however, the expression of COX-2 protein is the same as the expression of mRNA mentioned above, and CCEO has no inhibitory effect on COX-2 (Figure 6B). Studies have shown that LPS can jointly induce the expression of iNOS and COX-2 and cause the production of NO and PGE2, respectively [20]. However, during the chronic inflammation or resolution phase, the activities of the two may be separated [20]. Some of the literature shows that pretreatment of RAW264.7 cells with the antioxidant enzyme SOD can reduce the expression of LPS-stimulated iNOS but has no effect on COX-2 [21]. In the study of Koo et al., it was also found that the production of PGE2 decreased, but the performance of COX-2 protein did not change significantly [22]. It was deduced that it may be through the interference of the PGE2-degrading enzyme 15-PGDH (15-Hydroxyprostaglandin dehydrogenase) [23]; therefore, these results indicate that components of CCEO may inhibit the production of PGE2 independently of the COX-2 pathway.

### 2.6. Effects of CCEO on MAPK and NF-κB Pathway Expression

Since CCEO can reduce a number of LPS-induced inflammatory mediators, including pro-inflammatory cytokines (TNF-α, IL-6, and IL-1β), NO, and PGE2, and also inhibits the expression of iNOS, we further explored the upstream proteins that affect the inflammatory response. It is known that LPS can activate IKKα/β, leading to IκBα phosphorylation and subsequent ubiquitination and degradation, so that NF-κB is no longer restricted and can move into the nucleus, promoting the expression of various pro-inflammatory cytokines in macrophages [24]. MAPK pathway activation also promotes inflammatory responses, which has been confirmed in various inflammatory diseases and cancers [25]. Therefore, inhibition of the NF-κB and MAPK signaling pathways is considered to regulate inflammatory responses. The phosphorylation expression of three proteins in the MAPK pathway was detected. From Figure 7, it can be seen that after LPS stimulated RAW264.7 cells, the phosphorylation of ERK, p38, and JNK increased. After treatment with CCEO, it only significantly decreased at the highest concentration. The protein expression of phosphorylated ERK was measured. Next, we analyzed the performance of the NF-κB pathway. We first saw IKKα/β and IκBα. As shown in Figure 8, LPS stimulation tended to increase the phosphorylation of IKKα/β and also reduced IκBα due to degradation, but CCEO failed to inhibit IKKα/β phosphorylation. The performance of IκBα was significantly increased compared with the performance of the LPS induction group at a concentration of 5 µg/mL. However, there was a downward trend with the increase in CCEO concentration, which was opposite to the above-mentioned performance of CCEO in inhibiting pro-inflammatory factors and iNOS. CCEO also failed to effectively inhibit the nuclear translocation of NF-κB caused by LPS stimulation, and this result was not connected to the performance of IκBα (Figure 9). It is preliminarily inferred that CCEO can inhibit the expression of downstream inflammatory factors by reducing the activation of ERK protein, and low-concentration CCEO treatment can reduce IκBα degradation, but more evidence is still needed to confirm how CCEO affects the NF-κB pathway.

### 2.7. Effect of CCEO on the Expression of NRF_2_ Transcription Factor

NRF_2_ is responsible for maintaining cell tissue stability as well as redox balance. It can directly regulate the gene expression of HO-1, which reduces inflammation by inhibiting oxidative stress [13]. Given that the aforementioned results show that CCEO can reduce ROS content stimulated by LPS, we further explored whether it exerts antioxidant effects through the NRF_2_/HO-1 pathway (Figure 10). Figure 10A shows an upward trend in HO-1 expression with increasing CCEO concentration, which is significantly different from the induction group under 20 µg/mL CCEO treatment. In Figure 10B, it can also be found that in the group treated with CCEO, the expression of NRF2 located in the cytoplasm was lower, while the expression of NRF_2_ in the nucleus was increased compared with the group induced by LPS. The NRF_2_ pathway has been confirmed to reduce ROS production in macrophages by regulating antioxidant gene expression. The results show that LPS-induced ROS production was significantly inhibited by CCEO and has an HO-1 expression. Some reports in the literature indicate that increasing the activity of HO-1 can reduce the production of iNOS and NO, suggesting that CCEO may exert anti-inflammatory effects by activating the NRF_2_/HO-1 pathway and antioxidant effects.

## 3. Materials and Methods

### 3.1. Plant Materials

The leaves of *C. chinense* were collected in June 2023 in Ma-Ming Elementary School, Waipu District, Taichung City, Taiwan, and the collected samples were identified by Professor Yen-Hsueh Tseng (National Chung Hsin University). A voucher specimen (Y.S. Tseng 003, TCF) was deposited at the herbarium of the same university.

### 3.2. Preparation of CCEO and GC/MS Analysis

The 800 g dried leaves and 1.8 L double-distilled water were subjected to hydro-distillation in a Clevenger-type apparatus for 6 h, followed by the determination of oil contents. The CCLE was stored in airtight sample vials prior to analysis of gas chromatography–mass spectrometry (GC/MS) and bioactivity evaluation. An ITQ 900 mass spectrometer coupled with a DB-5MS column was used to analyze the composition of CCEO, and the temperature program was as follows: 45 °C for 3 min, then increased to 3 °C/min to 180 °C, and then increased to 10 °C/min to 280 °C and held for 3 min. The other parameters were injection temperature, 250 °C; ion source temperature, 200 °C; EI, 70 eV; carrier gas, He 1 mL/min; and mass scan range, 30–600 *m*/*z*. The volatile compounds were identified by Wiley/NBS Registry of mass spectral databases (V. 8.0, Hoboken, NJ, USA) and National Institute of Standards and Technology (NIST) Ver. 2.0 GC/MS libraries, and the Kovats indices were calculated for all volatile constituents using a homologous series of *n*-alkanes C9–C24. The major components were identified by co-injection with standards.

### 3.3. Unknown Compound Identification by High-Performance Liquid Chromatography (HPLC) and Nuclear Magnetic Resonance Spectroscopy (NMR)

A total of 0.5 g of essential oil was mixed with an equal amount of n-hexane. Solid phase extraction (SPE) was used to separate the compounds. The hexane and ethyl acetate fractions were analyzed using high-performance liquid chromatography (HPLC). For the hexane fraction, a solvent mixture of hexane and ethyl acetate (80:20) was used, and the analysis was run for 21 min. For the ethyl acetate fraction, hexane, ethyl acetate, and methylene chloride were used in a ratio of 95:2.5:2.5 and analyzed for 30 min. A Luna 5μ Silica column (Phenomenex) was used for separation, with detection at 254 nm.

To identify the compound structure, a Bruker AVANCEIII 400 NMR spectrometer was used at 300 K. The sample was redissolved in d-chloroform (CDCl3), and the chemical shifts of hydrogen (^1^H-NMR) and carbon atoms (^13^C-NMR) were measured in delta values (ppm). DEPT spectroscopy was used to detect primary, secondary, and tertiary carbons. Additionally, 2D NMR techniques like HSQC, HMBC, COSY, and NOESY were used to determine the structure of the three purified compounds.

### 3.4. Cell Culture and Cell Survival Assay

The macrophage cell line (RAW 264.7) was obtained from the American Type Culture Collection (ATCC). Cells were maintained in Dulbecco’s Modified Eagle Medium (DMEM, HyClone, Washington, DC, USA), supplemented with 10% fetal bovine sera (FBS, Gibco, Waltham, MA, USA), 1 mM sodium pyruvate (Corning, New York, NY, USA), 1% glutamax (Gibco), and 1% penicillin/streptomycin (Corning) at 37 °C in a humidified atmosphere of 5% CO_2_. Cells were subcultured at a density of 5 × 10^5^ cells/mL. Stock solutions of samples in dimethyl sulfoxide (DMSO, Sigma-Aldrich, St. Louis, MO, USA) were stored in the dark at 4 °C. Appropriate dilutions were prepared on the day of the experiments. The final concentration of DMSO did not exceed 0.2% (*v*/*v*). Cell viability was measured using a colorimetric MTT assay. Cells were seeded in 96-well plates at a density of 1 × 10^5^ cells/well. After 24 h incubated with samples, the culture medium was removed and replaced with fresh medium containing 0.5 mg/mL of MTT (Sigma-Aldrich) for 1 h. The supernatant was removed and 100 µL DMSO was added to dissolve formazan. Absorption at 540 nm for each well was measured using an ELISA microplate reader (BioTeck Instruments, Winooski, VT, USA).
Cell viability (%) = (sample OD_570_/control OD_570_) × 100

### 3.5. Anti-Inflammatory Measurement

#### 3.5.1. Nitric Oxide (NO) Assay

RAW 264.7 cells were seeded in 96-well plates at a density of 1 × 10^5^ cells/well. Adhered cells were then incubated for 24 h with or without 1 μg/mL of lipopolysaccharide (LPS, Sigma-Aldrich) in the absence or presence of samples. Curcumin was used as a positive control group. The supernatant detected was NO. For the product of NO, the supernatants were mixed with an equal volume of Griess reagent (Sigma-Aldrich) and incubated for 15 min at room temperature. The nitrite concentration was measured at 540 nm using an ELISA microplate reader.
$$\mathrm{NO}\;\mathrm{inhibition}\;(\%)=(1-\frac{sample\;induced\;{OD}_{540}-sample\;control\;{OD}_{540}}{induced\;{OD}_{540}-control\;{OD}_{540}})\times 100$$

#### 3.5.2. Prostaglandin E2 (PGE2), TNF-α, IL-1β, and IL-6 Assays

RAW 264.7 cells were seeded in 12-well plates at a density of 4 × 10^5^ cells/well. Adhered cells were then incubated for 24 h with or without 1 μg/mL of lipopolysaccharide (LPS, Sigma-Aldrich) in the absence or presence of samples. The supernatants detected were PGE2 (Cayman Chemical #514010, Ann Arbor, MI, USA), TNF-α (VAL609, Novus, St. Charles, MO, USA), IL-1β (VAL610, Novus, USA), and IL-6 (VAL604, Novus, USA) by using an ELISA Kit according to the manufacturer’s instructions.

#### 3.5.3. Reactive Oxygen Species (ROS) Assay

RAW 264.7 cells were seeded at a density of 1 × 10^5^ cells/well in a fluorescence-compatible 96-well plate (PerkinElmer Isoplate™-96 F TC). The plate was incubated for 2 h to allow the cells to adhere. The LPS and CCLE were added to incubate for 24 h. After 24 h, the supernatant was removed, and the wells were washed once with PBS. Then, 30 μM 2′,7′-dichlorodihydrofluorescein diacetate was mixed with PBS, and 100 μL of this mixture was added to each well. After a 45 min incubation, the fluorescence intensity was measured using a fluorescence spectrophotometer with an excitation wavelength of 485 nm and an emission wavelength of 535 nm. Higher fluorescence intensity indicates a higher level of ROS.

### 3.6. RNA Isolation and Quantitative Reverse Transcription PCR (qPCR)

Raw cells were cultured in a 6 cm plate at a density of 1 × 10^6^ cells per well. After two hours of culture, the LPS and different concentrations of essential oil were added at the same time. After six hours of culture, total RNA was extracted. RNA extraction was performed following the Total RNA Purification kit procedure (GeneMark, Zhubei, Taiwan, TR01-150), with the resultant RNA stored at −80 °C. Subsequently, RNA concentration was determined using the NanoVue™ Plus Spectrophotometer (GE HealthCare, Chicago, IL, USA) at 260/280 nm. For cDNA synthesis, 200 ng of total RNA was reverse-transcribed using SuperSAMScript IV Reverse Transcriptase (GeneMark, Zhubei, Taiwan, GRT004L). The reverse transcription process was carried out in a PCR Thermal Cycler (Astec, Fukuoka, Japan, PC-818A) with the following cycling conditions: initial incubation at 50 °C for 5 min for primer annealing, followed by extension at 50 °C for 30 min. Finally, the enzyme was deactivated at 85 °C for 15 min and the resulting cDNA was stored at −20 °C.

The expression of target junction genes in differentiated raw cell monolayers was evaluated via qPCR, with primer sequences detailed in Table 2. For the PCR reactions, we utilized PowerUp™ SYBR™ Green Master Mix (Thermo Fisher Scientific, Waltham, MA, USA), with a mixture comprising 10 μL of the master mix, 1 μL each of 10 µM primers for the positive and negative strands of the target gene, and 1 μL of the cDNA sample. The total reaction volume was adjusted to 20 μL with sterilized distilled water. PCR amplification was performed using a StepOne™ Real-Time PCR System (Applied Biosystems, Waltham, MA, USA) with the following cycling conditions: initial denaturation at 95 °C for 10 min, followed by 40 cycles of denaturation at 95 °C for 15 s and annealing at 60 °C for 1 min. Subsequently, a melt curve stage was conducted by raising the temperature to 95 °C for 15 s and then lowering it to 60 °C for 1 min. Data analysis was conducted using StepOne Software v2.3 (Thermo Fisher Scientific, USA). β-actin served as the endogenous control gene, and the relative expression of the target gene was calculated as 2^−ΔΔCt^.

### 3.7. Protein Extraction and Western Blot Analysis

RAW264.7 cells were seeded into 6 cm cell culture dishes with a density of 5 × 10^6^ cell/dish. After an overnight incubation, cells were treated with LPS (1 µg/mL) or various doses of test samples. After the cells were treated with essential oil for 30 min, the protein was collected to detect the expression of NF-κB. After 45 min of treatment, the expression levels were collected of protein detection p38, JNK, IκBα, and IKK. After 1.5 h of treatment, the protein was collected to detect the expression of ERK. After 6 h of treatment, the expression levels were collected of protein detection Nrf-2. After 12 h of treatment, the expression levels were collected of protein detection HO-1. Then, after 18 h of treatment, the expression levels were collected of protein detection iNOS and COX-2. Cells were lysed by radio-immuno precipitation assay (RIPA) buffer (Pierce Biotechnology, Rockford, IL, USA). Protein concentrations in the lysates were determined by Bio-Rad protein assay reagent (Bio-Rad Laboratories, Hercules, CA, USA). In total, 100 µg/lane of protein samples was separated by 10% SDS-PAGE. Then, the separated proteins were transferred onto polyvinylidene fluoride (PVDF) membrane for overnight. After the transfer, PVDF membranes were blocked with 5% skimmed milk for 90 min. After the membranes were washed with TBST, the membranes were incubated with NF-κB, p38, JNK, IκBα, IKK, ERK, HO-1, and iNOS and COX-2 antibodies for overnight or 16 h and then incubated with either HRP-conjugated anti-rabbit or anti-mouse antibodies for 2 h. Immunoblots were developed with the enhanced chemi-luminescence (ECL) reagents (Advansta Inc., San Jose, CA, USA); images were captured by ChemiDoc XRS+ docking system (Bio-Rad laboratories), and the protein bands were quantified by using Image lab software (Bio-Rad laboratories https://www.bio-rad.com/en-tw/product/image-lab-software?ID=KRE6P5E8Z, accessed on 19 May 2021).

### 3.8. Statistical Analysis

Data were expressed as the mean ± SD of three independent experiments. Statistical analysis was performed using GraphPad Prism 10 for Windows (GraphPad Software, La Jolla, CA, USA, https://www.graphpad.com/, accessed on 19 May 2021). Statistical significance was scored by using one-way ANOVA followed by Dunnett’s multiple test. * *p* < 0.05, ** *p* < 0.01, and *** *p* < 0.001 were considered statistically significant from sample treatment groups vs. the LPS group. In addition, the phosphorylation performance of IκBα and the nuclear translocation performance of NF-κB and Nrf-2 were compared with two-way ANOVA and Dunnett’s multiple test.

## 4. Conclusions

CCEO identified the main components through GC/MS, chromatography, and spectroscopic techniques as 1,8-cineole (13.73%), santolina triene (13.53%), germacrene D (10.67%), among others. These were obtained by separating and purifying essential oils The three compounds are 1-acetoxy-2-(2-hydroxypropyl)-5-methylhex-3,5-dien, 1-acetoxy-isopyliden-hex-5-en-4-one, and chrysanthemyl acetate. The first two compounds are novel discoveries. This study confirms that CCEO has outstanding anti-inflammatory effects and can reduce the production of NO and ROS in inflammatory reactions, with the IC_50_ of NO inhibition concentration at 10.3 μg/mL. Through RT-qPCR and ELISA measurements, it was confirmed that CCEO can reduce the expression of pro-inflammatory cytokines such as TNF-α, IL-6, and IL-1β. Using Western blotting to analyze its mechanism of action, it was found that CCEO inhibits iNOS expression but has no effect on COX-2. Further analysis of upstream proteins revealed that *C. chinense* essential oil impacts ERK protein on the MAPK pathway and the NF-κB inhibitor IκBα, inhibiting the phosphorylation of both, but there is no corresponding reduction. NF-κB transcription factor enters the nucleus. However, there is no corresponding reduction in the NF-κB transcription factor entering the nucleus. Additionally, exploration of the antioxidant pathway showed that wormwood essential oil can promote the NRF2/HO-1 pathway in cells, increase HO-1 expression, and facilitate NRF2 entry into the nucleus, thereby activating the cell’s antioxidant capacity (Figure 11). This results in the inhibition of the inflammatory response.

## Figures and Tables

**Figure 1 plants-13-02506-f001:**
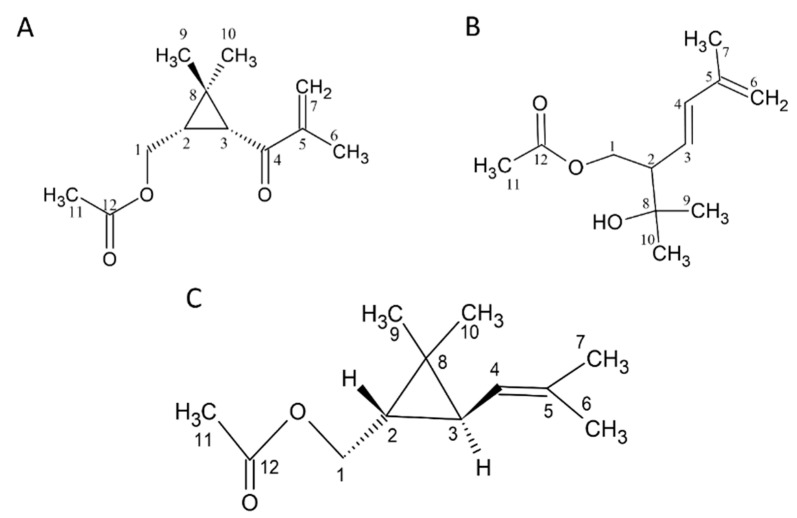
Structure of compounds **1** to **3**. (**A**) **1**: 1-acetoxy-(2R,3R)-isopyliden-hex-5-en-4-one, (**B**) **2**: (*E*)-1-acetoxy-2-(2-hydroxypropyl)-5-methylhex-3,5-diene, and (**C**) **3**: Chrysanthemyl acetate. The NMR spectra are shown in the Appendix A.

**Figure 2 plants-13-02506-f002:**
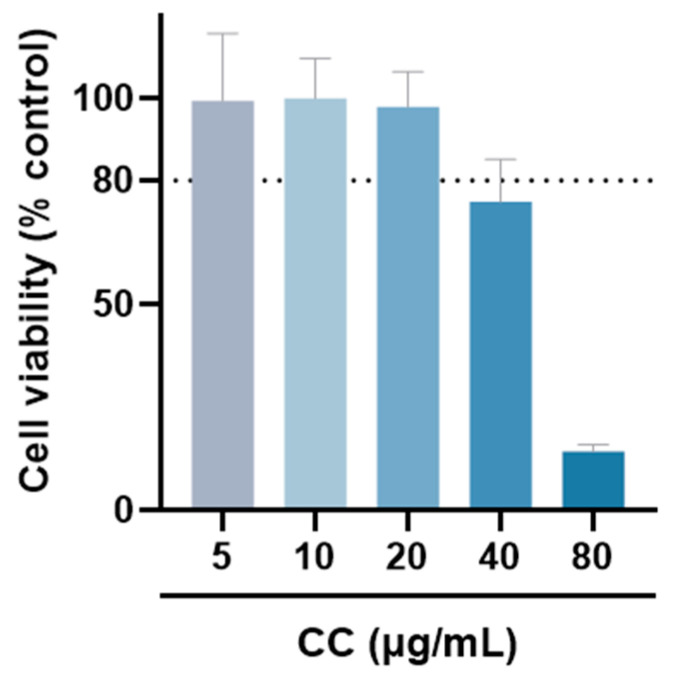
The cell viability of RAW264.7 treated with different concentrations of CCEO. Values are expressed as the mean ± SD (*n* ≥ 3). The dotted line represents the limit of 80% cell viability.

**Figure 3 plants-13-02506-f003:**
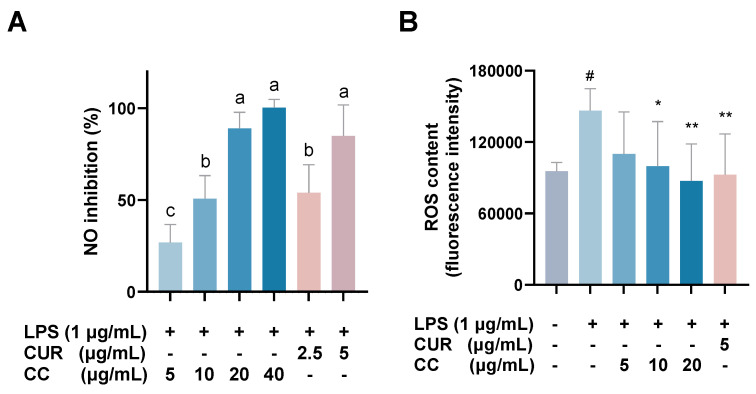
Effect of CCEO on (**A**) NO inhibition and (**B**) ROS content. RAW 264.7 cells were treated with CCEO (5–40 μg/mL) and curcumin (2.5, 5 μg/mL) with or without 1 μg/mL LPS for 24 h. NO inhibition was analysis by Griess assay. ROS content was analysis by DCFH-DA fluorescence. The data are represented as mean ± SD (*n* ≥ 3). A different letter indicates a significant difference among groups (*p* < 0.05). # *p* < 0.05 compared to control and * *p* < 0.05 and ** *p* < 0.01 compared to LPS-alone treatment.

**Figure 4 plants-13-02506-f004:**
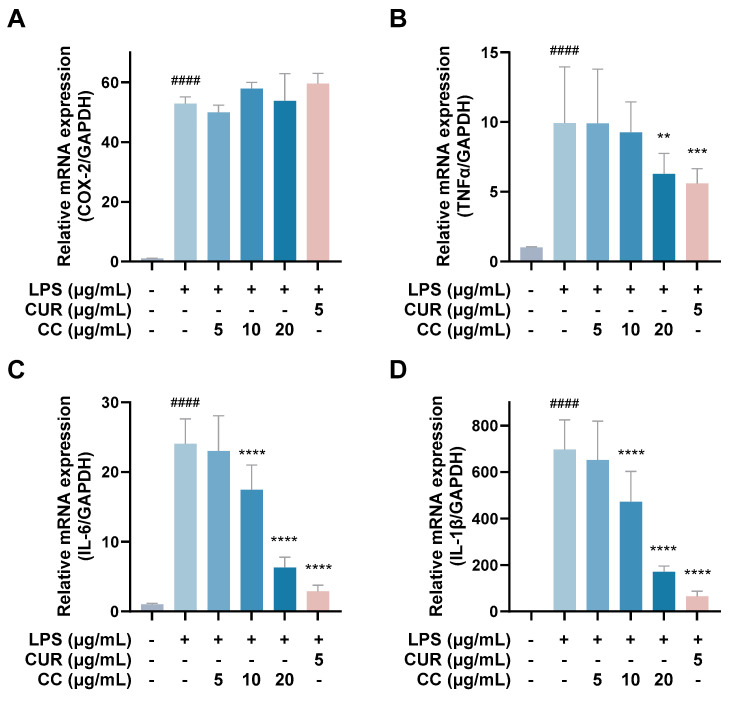
Effects of CCEO on the expression of COX-2 (**A**), TNF-α (**B**), IL-6 (**C**), and IL-1β (**D**) mRNA in RAW264.7 cells. RAW 264.7 cells were treated with CCEO (5–20 μg/mL) and curcumin (5 μg/mL) with or without 1 μg/mL LPS for 6 h. The mRNA expression was determined by quantitative RT-PCR. The data are represented as mean ± SD (*n* ≥ 3). #### *p* < 0.0001 compared to control and ** *p* < 0.01, *** *p* < 0.001, and **** *p* < 0.0001 compared to LPS-alone treatment.

**Figure 5 plants-13-02506-f005:**
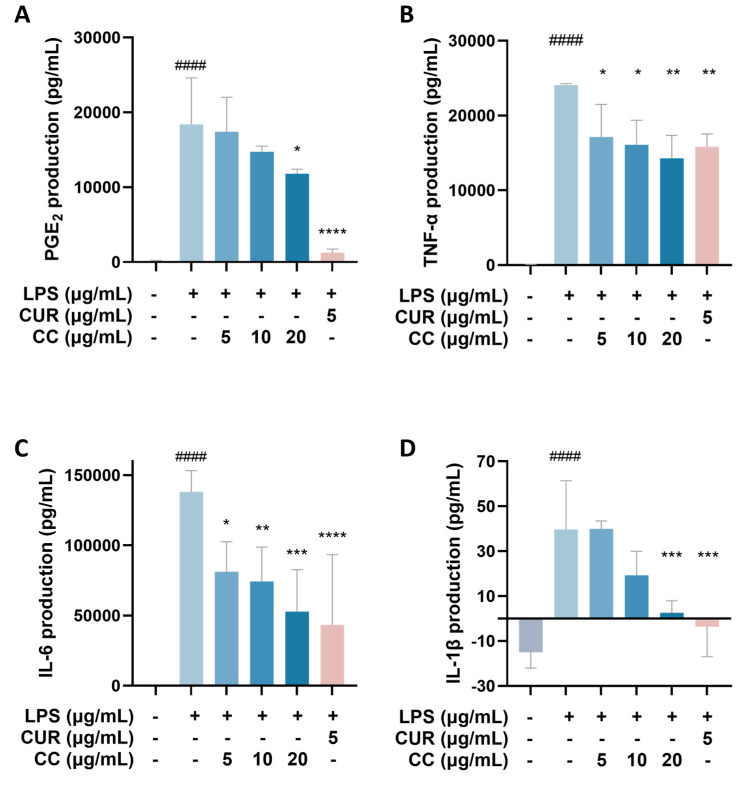
Effects of CCEO on the expression of PGE2 (**A**), TNF-α (**B**), IL-6 (**C**), and IL-1β (**D**) in RAW264.7 cells. RAW 264.7 cells were treated with CCEO (5–20 μg/mL) and curcumin (5 μg/mL) with or without 1 μg/mL LPS for 24 h. PGE2, TNF-α, IL-6, and IL-1β levels in the culture media were measured by commercially available assay kits. The data are represented as mean ± SD (*n* ≥ 3). #### *p* < 0.0001 compared to control and * *p* < 0.05, ** *p* < 0.01, *** *p* < 0.001, and **** *p* < 0.0001 compared to LPS-alone treatment.

**Figure 6 plants-13-02506-f006:**
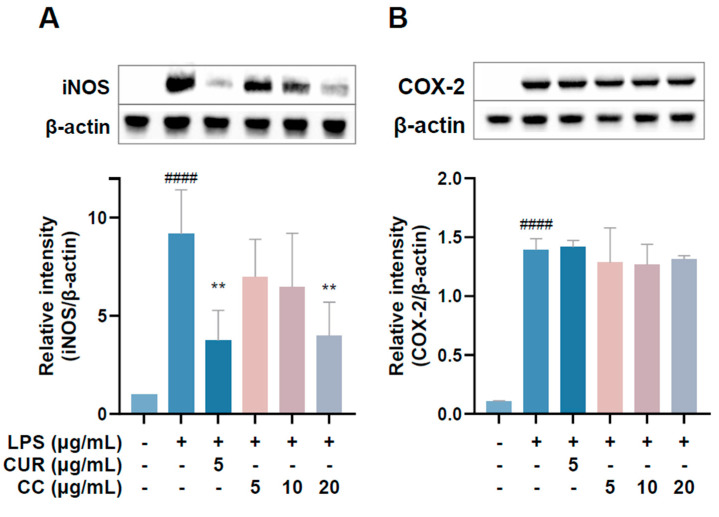
Effects of CCEO on the protein expression of iNOS. (**A**) COX-2 and (**B**) HO-1 in RAW264.7 cells. RAW 264.7 cells were treated with CCEO (5–20 μg/mL) and curcumin (5 μg/mL) with or without 1 μg/mL LPS. Protein samples were collected after 18 h for iNOS and COX-2 expression detection. Protein expression levels were determined by Western blot analysis with specific antibodies. β-actin served as an internal loading control. The data are represented as mean ± SD (*n* ≥ 3). #### *p* < 0.0001 compared to control and ** *p* < 0.01 compared to LPS-alone treatment.

**Figure 7 plants-13-02506-f007:**
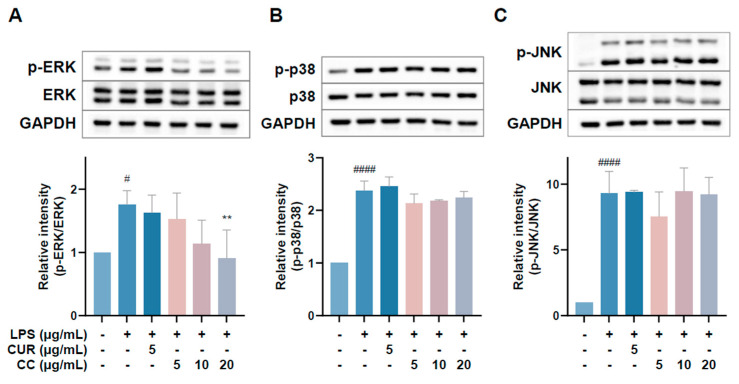
Effects of CCEO on phosphorylation of ERK (**A**), p38 (**B**), and JNK (**C**) in RAW264.7 cells. RAW 264.7 cells were treated with CCEO (5–20 μg/mL) and curcumin (5 μg/mL) with or without 1 μg/mL LPS. Protein samples were collected after 45 min of treatment for p38 and JNK expression detection and after 1.5 h for ERK expression detection. Protein expression levels were determined by Western blot analysis with specific antibodies. GAPDH served as an internal loading control. The data are represented as mean ± SD (*n* ≥ 3). # *p* < 0.05, #### *p* < 0.0001 compared to control, and ** *p* < 0.01 compared to LPS-alone treatment.

**Figure 8 plants-13-02506-f008:**
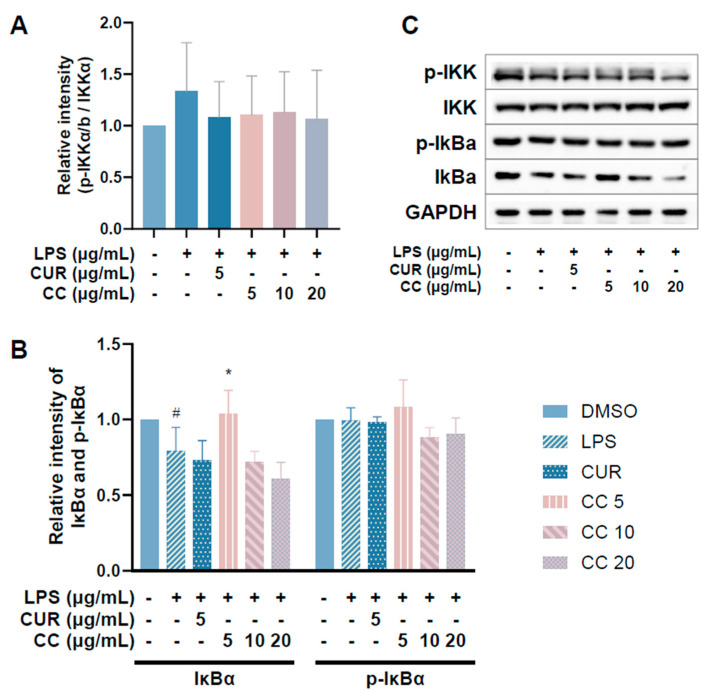
Effects of CCEO on phosphorylation of IKK (**A**) and IκBα (**B**) in RAW 264.7 cells. RAW 264.7 were treated with CCEO (5–20 μg/mL) and curcumin (5 μg/mL) with or without 1 μg/mL LPS. Protein samples were collected after 45 min of treatment and protein expression levels were determined by Western blot analysis with specific antibodies. GAPDH served as an internal loading control (**C**). The data are represented as mean ± SD (*n* ≥ 3). # *p* < 0.05 compared to control and * *p* < 0.05 compared to LPS-alone treatment.

**Figure 9 plants-13-02506-f009:**
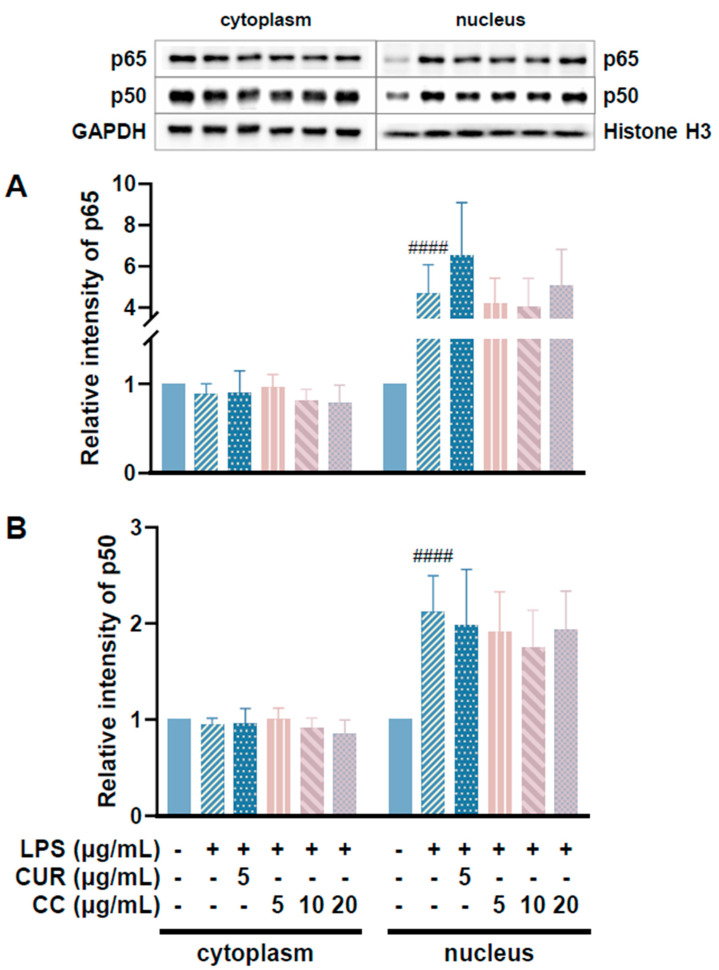
Effects of CCEO on expression of p65 (**A**) and p50 (**B**) in RAW264.7 cells. RAW 264.7 cells were treated with CCEO (5–20 μg/mL) and curcumin (5 μg/mL) with or without 1 μg/mL LPS. #### *p* < 0.0001 compared to control.

**Figure 10 plants-13-02506-f010:**
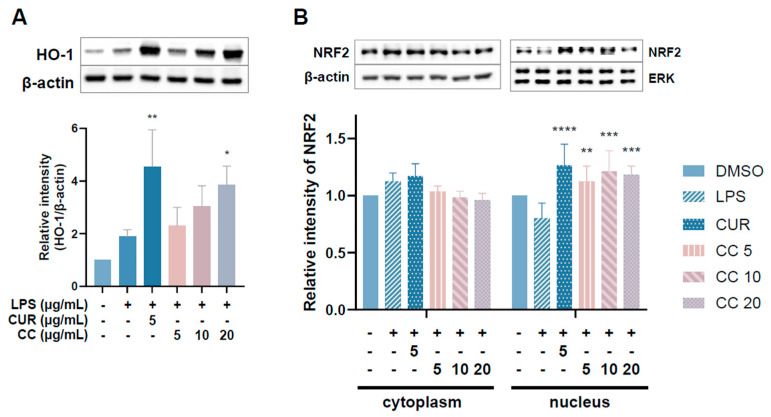
Effects of CCEO on phosphorylation of HO-1 (**A**) and NRF2 (**B**) in RAW264.7 cells. RAW 264.7 cells were treated with CCEO (5–20 μg/mL) and curcumin (5 μg/mL) with or without 1 μg/mL LPS. Protein samples were collected after 12 h of treatment for HO-1 expression detection and after 6 h for NRF2 expression detection. Protein expression levels were determined by Western blot analysis with specific antibodies. β-actin and ERK served as an internal loading control. The data are represented as mean ± SD (*n* ≥ 3). * *p* < 0.05, ** *p* < 0.01, *** *p* < 0.001, and **** *p* < 0.0001 compared to LPS-alone treatment.

**Figure 11 plants-13-02506-f011:**
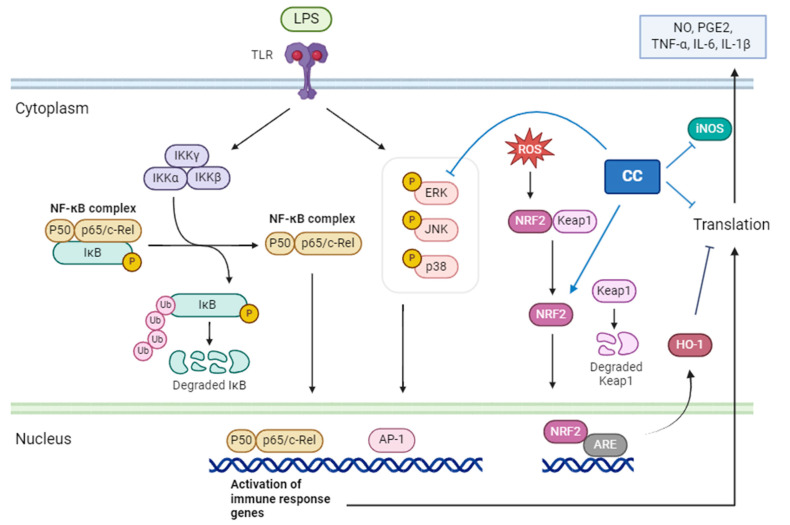
The mechanism of *C. chinense* essential oil inhibiting the RAW264.7 inflammatory response stimulated by LPS.

**Table 1 plants-13-02506-t001:** Composition analysis of *C. chinense* essential oil.

NAME	Concentration (%)	KI ^a^	Identification ^b^
Santolina triene	13.73	902	MS/KI/ST
*β*-Myrcene	0.78	996	MS/KI/ST
*p*-Cymene	0.72	1025	MS/KI/ST
Limonene	0.61	1030	MS/KI/ST
1,8-Cineole	13.53	1033	MS/KI/ST
*trans*-Chrysanthemol	1.61	1158	MS/KI/ST
Chrysanthemyl acetate	2.74	1268	MS/KI/ST
α-Copaene	0.98	1373	MS/KI/ST
1-Acetoxy-2-(2-hydroxypropyl)-5-methylhex-3,5-diene	0.23	1387	MS/ST/KI
1-Acetoxy-isopyliden-hex-5-en-4-one	2.6	1397	MS/ST/KI
*trans*-Caryophyllene	3.29	1416	MS/KI/ST
Allo-aromadendrene	0.62	1453	MS/KI
*α*-Gurjunene	0.81	1472	MS/KI
Germacrene D	10.67	1478	MS/KI/ST
*α*-Muurolene	1.02	1495	MS/KI/ST
*γ*-Cadinene	0.46	1509	MS/KI/ST
Cubebol	5.79	1515	MS/KI/ST
Calamenene	0.3	1518	MS/KI
Artedouglasia oxide C	3.69	1526	MS/KI
Lanciniata furanone F	0.67	1539	MS/KI
Germacrene B	3.6	1561	MS/KI/ST
Artedouglasia oxide D	2.09	1571	MS/KI
Caryophyllene oxide	1.41	1577	MS/KI/ST
*α*-Cadinol	3.54	1652	MS/KI/ST

^a^ Kovats index on DB-5MS column in reference to *n*-alkanes. ^b^ MS, NIST library and literature, KI, Kovats index, and ST, authentic standard compounds.

**Table 2 plants-13-02506-t002:** Primers used in QPCR.

Gene	Sequence
GAPDH	F: 5′-TCAACGGCACAGTCAAGG-3′ R: 5′-ACTCCACGACATACTCAGC-3′
COX-2	F: 5′-GCGACATACTCAAGCAGGAGCA-3′ R: 5′-AGTGGTAACCGCTCAGGTGTTG-3′
TNF-α	F: 5′-TAGCCAGGAGGGAGAACAGA-3′ R: 5′-TTTTCTGGAGGGAGATGTGG-3′
IL-6	F: 5′-CCGGAGAGGAGACTTCAC-3′ R: 5′-TCCACGATTTCCCAGAGA-3′
IL-1β	F: 5′-TTGAAGAAGAGCCCATCCTC-3′ R: 5′-CAGCTCATATGGGTCCGAC-3′

## Data Availability

All data generated or analyzed during this study are included in this published article and its Appendix A.

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
