# Peer review of "Composition Characterization of Crossostephium chinense Leaf Essential Oil and Its Anti-Inflammatory Activity Mechanisms"

_plants, 2024, doi:10.3390/plants13172506_

Round 1

Reviewer 1 Report

Comments and Suggestions for Authors

 In this study, authors extracted the essential oil of Crossostephium chinense (L.) Makino (Asteraceae) and analyzed EOs composition and anti-inflammatory activity. While there are some previous investigations of essential oil from this species, the novelty would be in the discovery of two new compounds and the study of the anti-inflammatory capacity of the EO. The introduction is very well written and provides a solid background for the present research. The rest of the manuscript is also okay, but some sentences are very hard to understand, so some English language editing is necessary. 

Here are some other comments on the manuscript.

1. When dealing with plant taxa that are not model organisms, one must always include the authority of the taxon, and at least in the introduction, introduce the plant taxon with its full name. Also, identification of the taxon is not enough; a voucher specimen must be deposited in the official herbarium collection, and, preferably, a scan of the voucher can be added to the supplemental for verification. 

2. There are a lot of formatting issues: wrong font, wrong spacing, but also in the naming of compounds. For example, n-butanol should be written as n-butanol. Names of the compounds contain a lot of spelling errors, i.e., iscaryophyllene is written as iso-caryophllene throughout the manuscript. Also, in Table 1, compound names should always start with the capital letter. 

3. Table 1. There is no need to include RT. The footnote labeled with * needs rephrasing since it does not have any meaning in English.

4. Section 2.3 Why were the results compared to Origanum and Cinnamomum, since these two species have very different chemical compositions and are not related at all to the studied species? What was the reasoning behind this comparison?

5. Section 3.3 needs to be rewritten to correspond to the style of writing of other sections. It reads like a recipe of what to do, not what was done.

6. Table 2. Guanins are written in lowercase letters in IL-6 primers. 

Comments on the Quality of English Language

Overall, the quality was good, but there were some issues with the English language in several paragraphs. See comments above. 

Author Response

Reviewer 1.

Q1. When dealing with plant taxa that are not model organisms, one must always include the authority of the taxon, and at least in the introduction, introduce the plant taxon with its full name. Also, identification of the taxon is not enough; a voucher specimen must be deposited in the official herbarium collection, and, preferably, a scan of the voucher can be added to the supplemental for verification. 

Response:

Thank you for your constructive feedback. We will include a description of Crossostephium chinense within the introduction section and highlight the corrections in yellow. We add the following description in the revised version, “The collected samples were identified by Professor Yen-Hsueh Tseng (National Chung Hsin University). A voucher specimen (Y.S. Tseng 003, TCF) has been deposited at the herbarium of the same university.

Q2. There are a lot of formatting issues: wrong font, wrong spacing, but also in the naming of compounds. For example, n-butanol should be written as n-butanol. Names of the compounds contain a lot of spelling errors, i.e., iscaryophyllene is written as iso-caryophllene throughout the manuscript. Also, in Table 1, compound names should always start with the capital letter. 

Response:

Thank you for your helpful advice. We greatly appreciate the reviewer's effort in correcting our formatting issues. In the article, we will adjust the font and spacing of the compounds as suggested. In Table 1, we have changed the names of all compounds to capital letters. All corrections will be highlighted in yellow.

Q3. Table 1. There is no need to include RT. The footnote labeled with * needs rephrasing since it does not have any meaning in English.

Response:

Thank you for your valuable advice. We will proceed with the necessary deletions and corrections.

Q4. Section 2.3 Why were the results compared to Origanum and Cinnamomum, since these two species have very different chemical compositions and are not related at all to the studied species? What was the reasoning behind this comparison?

Response:

Thank you for your valuable advice. Origanum and Cinnamomum are widely used essential oils in aromatherapy and natural medicine, known for their effectiveness in alleviating symptoms of inflammation and pain, such as arthritis. However, our research indicates that CCEO can be effective at relatively low concentrations and requires less dilution. Therefore, we believe that CCEO has significant potential for further development and promotion.

Q5. Section 3.3 needs to be rewritten to correspond to the style of writing of other sections. It reads like a recipe of what to do, not what was done.

Response:

Thank you for your valuable advice. We will simplify the description to improve the flow of the content and will highlight the changes in yellow in Section 3.3.

Q6. Table 2. Guanins are written in lowercase letters in IL-6 primers. 

Response:

Thank you for bringing this to our attention. We will make the necessary corrections and highlight them in yellow.

Reviewer 2 Report

Comments and Suggestions for Authors

ADDITIONAL COMMENTS

Composition Characterization of Crossostephium chinense Leaf Essential Oil and its Anti-inflammatory Activity Mechanism

The manuscript submitted for review represents a large-scale study of the essential oil of Crossostephium chinense (C. chinense). Here the research can be divided into main points - namely in the identification of the components of the essential oil and the second is the use of methodologies with which they establish the beneficial benefits for human health. For this, the review is divided in terms of chemical composition, activity, beneficial aspects of the article and its disadvantages.

Main results

1. Chemical composition of the essential oil:

Ø  Major components in C. chinense essential oil are 1,8-cineole (13.73%), santolina triene (13.53%), and germacrene D (10.67%).

Ø  Additionally, the authors identified three new compounds as 1-acetoxy-2-(2-hydroxypropyl)-5-methylhex-3,5-diene, 1-acetoxy-isopylidene-hex-5-en-4-one, and chrysanthemyl acetate.

2. Biological activity:

Regarding the biological activity, the authors make a large-scale study and it is expressed in the following experiments:

Ø  effectively inhibits the production of NO and ROS with an IC50 value of 10.3 µg/mL. This shows that the oil has a lot of antioxidant activity

Ø  reduces the expression of pro-inflammatory cytokines (TNF-α, IL-6, and IL-1β) in RAW264.7 macrophages.

Ø  Investigate the mechanisms of inhibition of iNOS expression and regulation of the inflammatory response

Useful aspects of the article

Ø  Methodology: They use analytical techniques (GC-MS, HPLC, NMR), which ensures high accuracy of the obtained results.

Ø  Integration of results: The authors have very well correlated the assays with in vitro biological activity

Ø  Novelty: The identification of new compounds in the essential oil and the in vitro biological activity make a significant contribution both to the field of phytochemistry and to the future application in medicine.

In my criticism, I will not overlook the weak points of the article

Weaknesses of the article

Ø  No evaluation regarding in vivo studies: Such laboratories cannot always be accessed. From my point of view, this does not affect the scientific value of the article.

Ø  Lack of comparison with other plants: In recent years, there has been a lot of research on essential oils. Here the authors can make a comparison with essential oils of the same family. They can make corelation both chemical composition and biological activity. In this context, they can make a correlation and find out which components are responsible for the high activity (QSAR).

The article provides valuable information on the chemical composition and anti-inflammatory properties of C. chinense essential oil. These studies will be useful, both for medicine and healthcare, and for understanding its safety and effectiveness in future clinical trials.

I give my positive rating.

I noticed an inaccuracy in the article:

In section 2.3. Anti-Inflammatory Activity of CCEO

The authors use the MTT method to comment on Anti-Inflammatory Activity. Here I must emphasize that the MTT method is most often used to evaluate cytotoxicity. COX-1 and COX-2 are the most preferred methods for evaluating anti-inflammatory activity. Can the authors provide more information - what is its relation to Anti-Inflammatory Activity?

This section deals more with antioxidant activity. It is known that the production of ROS and NO increases during the inflammatory process. Here the authors want to make a connection between the two processes (inhibition of NO production and Anti-Inflammatory Activity). They are interrelated processes. But in the text this idea is not perceived.

I recommend this part to edit.

Author Response

Reviewer 2.

The manuscript submitted for review represents a large-scale study of the essential oil of Crossostephium chinense (C. chinense). Here the research can be divided into main points - namely in the identification of the components of the essential oil and the second is the use of methodologies with which they establish the beneficial benefits for human health. For this, the review is divided in terms of chemical composition, activity, beneficial aspects of the article and its disadvantages.

Main results

  1. Chemical composition of the essential oil:

Ø  Major components in C. chinense essential oil are 1,8-cineole (13.73%), santolina triene (13.53%), and germacrene D (10.67%).

Ø  Additionally, the authors identified three new compounds as 1-acetoxy-2-(2-hydroxypropyl)-5-methylhex-3,5-diene, 1-acetoxy-isopylidene-hex-5-en-4-one, and chrysanthemyl acetate.

  1. Biological activity:

Regarding the biological activity, the authors make a large-scale study and it is expressed in the following experiments:

Ø  effectively inhibits the production of NO and ROS with an IC50 value of 10.3 µg/mL. This shows that the oil has a lot of antioxidant activity

Ø  reduces the expression of pro-inflammatory cytokines (TNF-α, IL-6, and IL-1β) in RAW264.7 macrophages.

Ø  Investigate the mechanisms of inhibition of iNOS expression and regulation of the inflammatory response

Useful aspects of the article

Ø  Methodology: They use analytical techniques (GC-MS, HPLC, NMR), which ensures high accuracy of the obtained results.

Ø  Integration of results: The authors have very well correlated the assays with in vitro biological activity

Ø  Novelty: The identification of new compounds in the essential oil and the in vitro biological activity make a significant contribution both to the field of phytochemistry and to the future application in medicine.

In my criticism, I will not overlook the weak points of the article

Weaknesses of the article

Ø  No evaluation regarding in vivo studies: Such laboratories cannot always be accessed. From my point of view, this does not affect the scientific value of the article.

Ø  Lack of comparison with other plants: In recent years, there has been a lot of research on essential oils. Here the authors can make a comparison with essential oils of the same family. They can make correlation both chemical composition and biological activity. In this context, they can make a correlation and find out which components are responsible for the high activity (QSAR).

The article provides valuable information on the chemical composition and anti-inflammatory properties of C. chinense essential oil. These studies will be useful, both for medicine and healthcare, and for understanding its safety and effectiveness in future clinical trials.

I give my positive rating.

Response:

Thank you very much for providing a highly positive evaluation of our article. We appreciate your feedback and have noted the two shortcomings you identified:

  1. The absence of in vivo studies.
  2. The lack of comparisons with other plants.

To address these points, we plan to conduct further experiments, focusing particularly on essential oils from other plants within the same family. Our aim is to correlate chemical composition with biological activity, identify the components that contribute to high activity (QSAR), and provide a more comprehensive report.

Q1. In section 2.3. Anti-Inflammatory Activity of CCEO

The authors use the MTT method to comment on Anti-Inflammatory Activity. Here I must emphasize that the MTT method is most often used to evaluate cytotoxicity. COX-1 and COX-2 are the most preferred methods for evaluating anti-inflammatory activity. Can the authors provide more information - what is its relation to Anti-Inflammatory Activity?

Response:

Thank you for your valuable feedback. We apologize for any misunderstandings that may have arisen from the presentation of our writing. We will revise the text to clarify that the MTT assay is used specifically for evaluating cytotoxicity, and these changes will be highlighted in yellow in Section 2.3.

COX-1 and COX-2 are key biomarkers for assessing anti-inflammatory activity, as these enzymes are pivotal in inflammatory responses, and their inhibition is commonly used as a measure of anti-inflammatory potential. Our study, based on western blotting results, demonstrates that CCEO inhibits the expression of iNOS without affecting COX-2. Instead, CCEO influences the ERK protein in the MAPK pathway and the NF-κB inhibitor IκBα by inhibiting their phosphorylation. Additionally, CCEO activates the NRF2/HO-1 pathway within cells, enhancing HO-1 expression and promoting NRF2’s nuclear translocation, which activates cellular antioxidant mechanisms to suppress inflammatory responses. Consequently, the relationship between COX-1, COX-2, and anti-inflammatory effects was not discussed in the conclusion.

Q2. This section deals more with antioxidant activity. It is known that the production of ROS and NO increases during the inflammatory process. Here the authors want to make a connection between the two processes (inhibition of NO production and Anti-Inflammatory Activity). They are interrelated processes. But in the text this idea is not perceived.

Response:

Thank you for your insightful question. While our initial tests indicated that CCEO can reduce the production of NO and ROS during the inflammatory response, further validation through RT-qPCR and ELISA measurements confirmed that CCEO also decreases pro-inflammatory cytokines, including TNF-α, IL-6, and IL-1β. Subsequent protein analysis revealed that CCEO activates the NRF2/HO-1 pathway within cells, leading to increased HO-1 expression and facilitating NRF2 translocation into the nucleus, thereby enhancing the cell’s antioxidant capacity and suppressing the inflammatory response. As a result, we did not elaborate on the specific mechanisms by which CCEO inhibits NO and ROS.

Reviewer 3 Report

Comments and Suggestions for Authors

The manuscript titled “Composition Characterization of Crossostephium chinense Leaf Essential Oil and its Anti-inflammatory Activity Mechanism” submitted by, Ming-jie Han and collaborators is very well described.

On this research article the authors studied the composition characteristics and anti-inflammatory activity mechanisms of the essential oil from the leaves of Crossostephium chinense commonly known as "Qing Ai" which is a perennial herb commonly found in East Asia, traditionally used to treat various ailments. The chemical composition of the essential oil was determined by using gas chromatography and a couple compounds were discovered. The anti-inflammatory activity of the essential oil was studied by lipopolysaccharides inducing macrophages inflammation finding as results an effectively inhibition the production of NO and ROS and  a significant reduction of the expression of pro-inflammatory cytokines such as TNF-

Author Response

Reviewer 3

The manuscript titled “Composition Characterization of Crossostephium chinense Leaf Essential Oil and its Anti-inflammatory Activity Mechanism” submitted by, Ming-jie Han and collaborators is very well described.

On this research article the authors studied the composition characteristics and anti-inflammatory activity mechanisms of the essential oil from the leaves of Crossostephium chinense commonly known as "Qing Ai" which is a perennial herb commonly found in East Asia, traditionally used to treat various ailments. The chemical composition of the essential oil was determined by using gas chromatography and a couple compounds were discovered. The anti-inflammatory activity of the essential oil was studied by lipopolysaccharides inducing macrophages inflammation finding as results an effectively inhibition the production of NO and ROS and a significant reduction of the expression of pro-inflammatory cytokines such as TNF-

Respomnse

Thank you for your review and affirmation.
